# Integrating Landscape Ecological Risks and Ecosystem Service Values into the Ecological Security Pattern Identification of Wuhan Urban Agglomeration

**DOI:** 10.3390/ijerph20042792

**Published:** 2023-02-04

**Authors:** Haojun Xiong, Haozhi Hu, Pingyang Han, Min Wang

**Affiliations:** 1College of Horticulture & Forestry Sciences, Huazhong Agricultural University, Wuhan 430070, China; 2Key Laboratory of Urban Agriculture in Central China, Ministry of Agriculture and Rural Affairs, Wuhan 430070, China

**Keywords:** ESV, LER, ESP, minimum cumulative resistance model, WUA

## Abstract

Urban agglomerations are the main form of China’s future promotion of new urbanization development. Nevertheless, their accelerated expansion and development are increasingly threatening the security of regional ecosystems. The identification and optimization of ecological safety patterns (ESPs) is the fundamental spatial way to guarantee the ecological safety of urban circles and realize the sustainable development of the socio-economic and ecological environment. Nevertheless, from the perspective of urban green, low-carbon, and ecological restoration, regional safety evaluation still lacks a complete framework integrating ecological elements and social and natural indicators. Moreover, the evaluation method of ESPs also has a lack of judgment on the long-term change dynamics of regional landscape ecological risks and ecosystem service values. Thus, we proposed a new regional ecological security evaluation system based on ecosystem service value (ESV) and landscape ecological risk (LER), using the Wuhan urban agglomeration (WUA) as the research object. This study analyzed LER and ESV’s spatial and temporal changes over nearly 40 years from 1980 to 2020. LER and LSV were used as ecological elements combined with natural and human-social elements to jointly model the resistance surface of the landscape pattern. Applying the minimum cumulative resistance model (MCR), we identified green ecological corridors, constructed the ESPs of WUA, and proposed optimization measures. Our results show that: (1) The proportion of higher- and high-ecological-risk areas in WUA has decreased from 19.30% to 13.51% over the past 40 years. Over time, a “low–high–low” hierarchical distribution characteristic centered on Wuhan city was gradually formed in the east, south, and north; the total value of ecosystem services increased from CNY1110.998 billion to CNY1160.698 billion. The ESV was higher in the northeastern, southern, and central parts of the area. (2) This study selected 30 ecological source areas with a total area of about 14,374 km^2^ and constructed and identified 24 ecological corridors and 42 ecological nodes, forming a multi-level ecological network optimization pattern with intertwined points, lines, and surfaces, increasing the connectivity of the ecological network and improving the ecological security level of the study area to a large extent, which is of great significance to promote the ecological priority and green-rise strategy of WUA and the high-quality development path of the green ecological shelter.

## 1. Introduction

Over the past 40 years, with accelerated urbanization and rapid population growth, the change in land use and occupation by humans has induced a series of ecological degradation problems [1,2,3,4]. In addition, it has caused landscape fragmentation and degradation of functional ecosystem services, which has posed an extremely serious threat to urban ecological security and regional sustainable development [5]. Therefore, analyzing the spatial and temporal changes of LER and ESV and optimizing the ecological security pattern during urbanization and construction is a meaningful way to safeguard the function of urban ecosystems, improve the quality of the ecological environment, and achieve regional sustainable development [6,7].

LER assessment is an essential subfield of ecological risk evaluation at the regional scale [8]. LER mainly relies on the theory of landscape ecology, coupling landscape patterns and ecological processes, which focus on the spatial and temporal heterogeneity of risks and the adverse consequences of scale effects [9]. The evaluation objects of LER mainly consist of watersheds [10], administrative regions [11], urban areas [12], and other regional landscapes. The evaluation methods mainly include the risk “source–sink” method and the landscape index method. The landscape risk index based on land use type is more commonly applied in the assessment of LER by Land Use/Cover Change (LUCC) [13].

ESV refers to the benefits that humans receive directly or indirectly from ecosystems. It plays a critical role in human well-being, health, and the interdependence of humans and all life on earth [14,15,16,17,18]. Since the United Nations launched the ecosystem assessment project globally in 2003 [19,20], ESV has become the focus of research on land use change and ecosystem interactions. The current ESV assessment methods mainly include the equivalent factor assessment method [21] and the functional value assessment method [22]. ESV assessment contributes to understanding the quality and changes of ecosystems, providing a reliable basis for ecological compensation and better planning of ESPs [23,24,25].

ESP aims to provide healthy and sustainable conditions for the development of ecosystem services by maintaining and controlling ecological processes [26], which promote the coordination of regional ecological and environmental protection with economic development. The construction of ESPs is beneficial for managing ecological risks in the landscape and maintaining the integrity and stability of ecosystem service functions [27]. The concept of ESP can be traced back to the 1990s as a development of the study of landscape security patterns at the regional scale, which has adapted to the development needs of biological conservation and ecological restoration research [28]. In recent years, the research objects of ESP have been mainly economic development areas [29], water connotation areas [30], and ecologically fragile areas [31]. In addition, the spatial scale of research is also diversified, including the macro-regulation of various natural resources and the micro-regulation of special geographical conditions of each local and administrative unit [32,33]. The three critical components of “ecological source site identification, resistance surface creation, and corridor extraction” [34] have gradually become the core elements of ecological security pattern construction, which has initially formed a construction paradigm. An ecological source is a site with good ecological function, ecological quality, and rich biological resources that is characterized by stability and expansion [35,36]. The identification methods of ecological source areas are mainly existing ecological reserves and the construction of ecologically integrated index systems [37,38]. The creation of ecological resistance surfaces is the basis for corridor extraction. In this regard, the “minimum path” between ecological source areas is analyzed based on the minimum resistance model (MCR) theory [39,40], which benefits biological migration and dispersal. The construction of the resistance surface is mainly based on natural and anthropogenic disturbance factors. Nevertheless, this index system does not adequately consider the regional ecological processes and ecological needs. In fact, ecological resistance coefficients are differentiated by the dynamics of landscape fragmentation [41] and land use forms [42] over time, as well as by the interaction of land use and ecological processes. Especially in rapidly urbanizing regions, especially in urban areas, rapid urban expansion and rapid population growth have caused intricate and drastic changes in land use and ecological processes, and there are significant human–land conflicts and ecological security risks. To address the ecological vulnerability and development uniqueness of rapidly urbanizing regions, carrying out an assessment based on ESPs also needs to focus on changes in landscape ecological risks and ecological processes over time, providing a basis for scientifically proposing refined policies and recommendations. Yet, in the background of the current era of green, low-carbon, and ecological restoration, the ESPs constructed by integrating ESV and LER can meet the dual needs of improving ESV and reducing LER to serve the arrangement and planning of urban territorial spatial strategy more reasonably and sustainably.

WUA is located in the middle and lower reaches of the Yangtze River, which is an important engine to pull the rise of central China and a national comprehensive supporting reform pilot area. Recently, the WUA Development Plan has been approved by the National Development and Reform Commission, becoming the seventh metropolitan area plan. Over the past four decades, WUA has been undergoing urban–rural development transformation. The ecosystem function and ecological pattern have been seriously disturbed, with ecological pollution problems such as lake filling, occupation, and water pollution. To improve the ecosystem stability of WUA to a greater extent and promote the region’s sustainable development, this study aims to clarify the spatial and temporal changes of LER and ESV and propose ecological control of WUA. At the same time, both LER and ESV as ecological elements are combined with natural and social elements to build an ESP evaluation system and explore the optimization strategy of ESPs in WUA to guide the “green city agglomeration” of ecological synergy development.

## 2. Materials and Methods

### 2.1. Study Area

WUA is located in the eastern part of Hubei province and the middle reaches of the Yangtze River (29.2′~31.51′ N, 112.30′~116.7′ E, Figure 1), with an area of 57,800 km^2^ and a resident population of 31,894,000 [43]. WUA is an urban association composed of Wuhan and eight large- and medium-sized cities within a radius of about 100 km around Huangshi, Ezhou, Huanggang, Xiaogan, Xianning, Xiantao, Qianjiang, and Tianmen. It is the core region of economic development in Hubei and an essential strategic pivot point for the rise of central China. The urban agglomeration has a dense network of rivers and diverse types of landforms, mainly plains, rivers, hills, and mountains. In recent years, with the expansion of the urbanization process, the contradiction between population, economy, and land resources has become increasingly acute. The problem of environmental pollution in WUA has gradually come to the fore, resulting in a series of ecological security problems such as landscape fragmentation and degradation of ecosystem service functions [44]. Therefore, it is urgent to delineate the ecological protection space, build green ecological corridors, and construct a new ecological security pattern for WUA with water and greenery blending and complementing diversity.

### 2.2. Data Sources

The research data in this paper include land use data, basic geographic data, vegetation cover data, and DEM data. The five-year land use data and vegetation cover data of WUA for 1980, 1990, 2000, 2010, and 2020 were obtained from the Resource Science Data Center of the Chinese Academy of Sciences (http://www.resdc.cn) (accessed on 18 March 2022). The DEM data came from the Resource and Environment Data Cloud Platform, Institute of Geographical Sciences and Resources, Chinese Academy of Sciences. The five-year land use spatial resolution is 30 m, and the accuracy rate of each land type is over 90% [45]. The land use types were reclassified into six categories: forest land, grassland, water, arable land, construction land, and unused land, taking into account the situation of the study area. The data vector boundaries, river network, and other basic geographic information data were obtained from the National Geographic Information Resources Catalogue Service System (https://www.webmap.cn/) (accessed on 5 April 2022). The DEM elevation data with 30 m resolution were obtained from the geospatial data cloud platform.

According to the relevant landscape ecology studies, the landscape evaluation sample area was generally 2 to 5 times the patch area [46]. Based on the area of the study area, this study divided it into 3 km × 3 km evaluation cells, with a total of 6797 cells. The ecological risk index and ecosystem service value were calculated separately for each evaluation cell.

### 2.3. Research Framework

The research framework can be divided into four steps, as shown in Figure 2. The first step was to assess the spatial and temporal changes of LER and ESV in the study area over the past 40 years and provide spatially differentiated and refined ecological control suggestions for optimizing the safety pattern. Secondly, a resistance surface model of the landscape pattern was constructed using three elements of “ecology–nature–society”, and the indicator weights were determined using principal component analysis. Third, by integrating the ecological sources and landscape pattern resistance surface, the MCR model was run to extract potential ecological corridors. Finally, the integrated consideration of ecological source sites, key potential corridors, and ecological control was proposed to construct and optimize the ESPs. The following sections describe each research process in detail.

### 2.4. Research Method

#### 2.4.1. Assessment of Landscape Ecological Risk

The landscape loss index ($R_{i}$) and the proportion of each type of land area in the evaluation unit are used as assessment indices to establish the landscape ecological risk index model (ERI, ${\mathrm{ERI}}_{k}$), which describes the overall landscape ecological risk of the study area [47]. The calculation formula is as follows:(1)$${\mathrm{ERI}}_{k}=\overset{n}{\underset{i=1}{\sum }}\frac{A_{\mathrm{ki}}}{A_{k}}R_{i}$$
where ${\mathrm{ERI}}_{k}$ represents the ecological risk index of the kth ecological risk plot; $A_{\mathrm{ki}}$ is the area of landscape type i within the kth risk plot; $A_{k}$ is the area of the kth risk plot; $R_{i}$ is the landscape loss index; and n is the number of landscape types.
(1)The landscape loss index (
$R_{i}$)

$R_{i}$ indicates the result of the loss of external disturbances acting on the landscape interacting with its own vulnerability [48]. The calculation formula is as follows:(2)$$R_{i}=E_{i}\times F_{i}$$
where $E_{i}$ is the landscape disturbance index, $F_{i}$ is the landscape vulnerability index.
(2)The landscape obstruction index (
$E_{i}$
)

In this paper, the landscape fragmentation index $(C_{i}),$ landscape splitting index $(S_{i})\;,$ and landscape dominance index $\left(D_{i}\right)$ are used to construct the landscape obstruction index $\left(E_{i}\right)$, which is calculated as follows.
(3)$$E_{i}={\mathrm{aC}}_{i}+{\mathrm{bS}}_{i}+{\mathrm{cD}}_{i}$$

① The landscape fragmentation index $C_{i}$ is caused by natural or human factors. Continuous and homogeneous patches in the landscape gradually transform into discontinuous and separated patches. The larger the fragmentation value, the lower the internal stability of the landscape unit. The calculation formula is as follows:(4)$$C_{i}=\frac{n_{i}}{A_{i}}$$
where $n_{i}$ is the number of the first landscape patches and $A_{i}$ is the total area of the first landscape.

② The landscape splitting index $S_{i}$ refers to the dispersion of individual patches within the overall landscape area and the degree of diffusion. The larger the value of separation, the more dispersed the patches are in the landscape distribution, the higher the fragmentation of the landscape, and the greater the degree of human interference. The calculation formula is as follows:(5)$$S_{i}=\frac{1}{2}\sqrt{\frac{n_{i}}{A}}\frac{A}{A_{i}}$$
where A  is the total landscape area of the study area, $n_{i}$ is the number of patches of the ith landscape, and $A_{i}$ is the total area of the ith species.

③ The landscape dominance index $D_{i}$ is used to characterize the dominant component of a landscape within the overall area. The calculation formula is as follows:(6)$$D_{i}=\frac{\left(Q_{i}+M_{i}\right)+2L_{i}}{4}$$
where $L_{i}$ is the ratio of the area of the ith landscape to the total area of the study area, $M_{i}$ is the ratio of the number of the ith landscape patches to the total number of patches, and $Q_{i}$ is the ratio of the number of the ith landscape plot to the total number of risk plots.

Additionally, a,b,c are the weights of each landscape index and a+b+c=1. According to the actual situation of the study area and literature research, the fragmentation index $C_{i}$ is considered to be the most important, followed by the separation index, and finally the dominance index, and the three indices are assigned values of 0.5, 0.3, and 0.2, respectively [49,50].
(3)The landscape vulnerability index $\left(F_{i}\right)$.

The landscape vulnerability index $F_{i}$ indicates the degree of vulnerability of various landscapes after external disturbance [51], and the more significant the vulnerability value, the greater the ecological risk. According to the previous experience and the characteristics of WUA [52,53], the vulnerability of the six landscape types was assigned from high to low as follows: unused land 6, water 5, arable land 4, grassland 3, forest land 2, and construction land 1, which were normalized to obtain the vulnerability indices of each landscape type, 0.28, 0.24, 0.19, 0.14, 0.1, and 0.05, respectively.

#### 2.4.2. Assessment of the Ecosystem Service Value

Based on the revised ESV equivalent factor table by Xie et al. [54] and the research results of other scholars, the land use types in the study area were divided into six categories: unused land, water, cropland, grassland, forest land, and construction land, where the value factor was set to 0 based on the land use nature of construction land. At the same time, the food production per unit area of cropland and the reference price of food in WUA in 2018 were used as the basis. Moreover, the equivalent factor of ESV in WUA was corrected according to the rule that the economic value provided by the ecosystem per unit area is 1/7 of the food value provided by its arable land [55]. Through consulting the statistical yearbooks of nine cities in WUA, we obtained the unit area grain yield of WUA in 2018 as 6367.91 kg/hm^2^. We also obtained the 2018 grain purchase price of 1.26 CNY/kg by consulting the Hubei Provincial Grain Bureau. Finally, the value equivalent factor of WUA was obtained as 1146.22 CNY/hm^2^. In summary, the ESV per unit area of the WUA ecosystem is shown in Table 1.

#### 2.4.3. Optimization of Ecological Security Pattern

The Minimum Cumulative Resistance Model (MCR) is the least-cost pathway that reflects the cost of species crossing different landscape substrates from source sites. The resistance surface reflects the trend of species’ spatial movement [56]. The basic formula is as follows:(7)$$\mathrm{MCR}=f_{\min}\overset{i=m}{\underset{j=n}{\sum }}D_{\mathrm{ij}}\times R_{i}$$
where MCR is the minimum cumulative resistance value; f denotes the positive correlation between the minimum cumulative resistance model and ecological processes, which is an unknown monotonically increasing function; ∑ denotes the distance between raster i and source j across all cells and the accumulation of resistance; $D_{\mathrm{ij}}$ denotes the spatial distance (unit: m) of ecological elements from source j to landscape cell i; and $R_{\mathrm{ij}}$ denotes the resistance value of landscape cell i to the movement of a particular ecological element.

Ecological source identification

The ecological source is an area with high ecosystem service values and intact ecological patterns, with strong resistance to disturbance and ecological solid stability [57,58]. Considering the actual situation of WUA, forested areas with strong resistance to disturbance and water areas with functions such as water containment and runoff storage were selected as ecological source areas. The minimum area of the mountain ecological source site is 160 km^2^ or more, and the area of the water ecological source site is 70 km^2^ or more.

2.Landscape pattern resistance surface construction

Landscape pattern resistance is the resistance that needs to be overcome for the ecological and material flows of ecosystems to operate spatially [59,60]. The migration of species and the circulation of ecological functions are closely related to factors such as climate, vegetation, and human disturbance. In this study, the evaluation indices were set as two ecological factors, LER and ESV; four natural indicators, slope, elevation, distance from water bodies, and normalized difference vegetation index (NDVI); and three human and social indicators, distance from construction land, distance from roads, and population density. We selected nine indicators to characterize the ecological resistance of natural and anthropogenic elements. We divided these nine indicators into five levels, from level 1 to level 5, representing low, lower, medium, higher, and high resistance levels, respectively, with higher values representing higher ecological resistance and higher external interference (Table 2). After that, we used spatial principal component analysis to determine the contribution of each index to the ecological resistance and calculated the weight of each index according to the contribution [61].

3.Ecological corridor extraction and node identification

The ecological corridor is the minimum resistance channel between two neighboring source sites, which is a portable channel for material circulation and energy flow in the ecosystem and a key element to improve the integrity of the regional ecosystem and enhance the close connection between ecological source sites [62]. According to the minimum cumulative resistance model, based on the constructed resistance surface, we used ArcGIS to calculate and simulate the minimum resistance between each ecological source site in the regional landscape pattern to complete the construction of ecological corridors.

Ecological nodes are the critical nodes in the ecosystem and the weak ecological function areas. Strengthening the construction and protection of ecological nodes has the advantage of improving the stability of the whole ecosystem structure [63]. In this study, we identified the intersection of the “ridge line” of the high resistance value distribution and the ecological corridor as the WUA ecological node in ArcGIS.

## 3. Results

### 3.1. Spatial and Temporal Changes in LER and ESV

#### 3.1.1. Spatial Differentiation of LER

We calculated the landscape pattern indices of different landscape landscapes in WUA from 1980 to 2020 with fragstats. Regarding land use status, farmland and forest land were the leading landscape types in WUA. The cropland area was on a decreasing trend, but the number of cropland patches was increasing yearly. The fragmentation and separation of farmland landscapes were increasing year by year, leading to a slow increase in the loss index of farmland and an increase in the ecological risk of farmland landscapes year by year (Table 3). The forest area showed a trend of first decreasing and then increasing. The number of patches tended to increase year by year, making the fragmentation and separation of the forest landscape increase slowly year by year. The forest integrity has been damaged to some extent. With urbanization’s impact, land construction has expanded rapidly in the past 40 years. The number of its patches has mostly stayed the same, showing an increase and then a decrease. Overall, from the viewpoint of landscape types, the landscape loss index of cropland, forest, grassland, and watershed was low, and the landscape loss index varied in different periods. The landscape loss index of cropland, forest land, and grassland showed a slightly increasing trend. Since it was influenced by the excellent effect of the Grain for Green project policy and Returning Farmland to Lakes policy, the landscape loss index of the watershed initially showed a slight downward trend. Although the landscape loss degree index of unused land was higher, it was limited by its smaller area and has less impact on the overall ecological risk.

According to the ecological risk evaluation model, we evaluated the ecological risk evaluation results of 6797 ecological risk plots in 1980, 1990, 2000, 2010, and 2020 using the kriging interpolation method in ArcGIS 10.6. We found that the ecological risk evaluation values ranged from 0.0065 to 0.0523 for the five periods through data processing. The assessment results were classified into five levels with the natural interruption point grading method: low risk [0, 0.0100], lower risk (0.0100, 0.0126], medium risk (0.0126, 0.0153], higher risk (0.0153, 0.0196], high risk (0.0196, ∞] (Table 4). From 1980 to 2020, the higher-risk and high-risk areas decreased by 2.34 km^2^ and 3360.06 km^2^, while the lower-risk and medium-risk areas increased by 1444.86 km^2^ and 2379.06 km^2^, respectively, indicating that the overall landscape ecological risk in WUA has increased. The ecological risk level in the study area decreased spatially from the center to the periphery; the high-risk and higher-risk areas were mainly distributed in Wuhan city and its border areas with neighbouring cities, showing a cluster-like distribution and a characteristic of shrinking from the center to the periphery. The low-risk and lower-risk areas were mainly distributed in the northern part of Xiaogan City, the northern and northeastern part of Huanggang City, the southern part of Huangshi City, and Xianning City, with an expansion trend in the area. The risk level decreased in the urban centers of Wuhan City, Huangshi City, and Ezhou City (Figure 3). The spatial changes in the ecological risk level of the landscape indicated that WUA was working to prevent ecological risks in the core urban areas and focusing on ecological restoration. During 1990–2000, the urbanization process was slow; the medium-risk and low-risk areas increased, the high-risk areas decreased slightly, and the overall ecological risk mainly stayed the same. During 2000–2010, with the rapid urban development and construction in Wuhan since the new century, the intensity of human activities increased, and the area of high-risk areas and higher low-risk areas increased by 0.12%, posing a severe threat to regional ecological security (Figure 4). Another promising finding was that since the early 20th century, Hubei Province has vigorously promoted the implementation of the policy of returning fields to lakes and leveling floods. The ecological mitigation of the Yangtze River has improved year by year, effectively enhancing the ecological functions of the lakes and rivers and continuously reducing the regional ecological risks, which is of great significance to the construction of a significant ecological security barrier in the upper reaches of the Yangtze River.

#### 3.1.2. Analysis of Spatial and Temporal Variation in ESV

In 1980, 1990, 2000, 2010, and 2020, the total ESV in WUA was 111.0998, 1150.257, 1153.272, 1153.006, and CNY116.0698 billion, respectively, showing an overall upward development trend. The total ESV increased by 3.534%, 0.262%, −0.023%, and 0.667% from 1980–1990, 1990–2000, 2000–2010, and 2010–2020, respectively, with decreasing rates of ESV change from year to year. ESV supply in WUA was mainly dominated by forest land and waters, accounting for more than 70%. From 1980 to 2020, the ESV of waters rose by 137%, reflecting the firm implementation of the policy of returning fields and fisheries to lakes, spatial control of lakes, and restoring the ecological environment (Table 5). However, during this period, the ESV of forest land, grassland, and unused land decreased by CNY5.114, 0.655, and 0.224 billion indicating that the construction of high-speed economic-centered urban development created coercion to form duress on the regional ecological environment and even cause ecological damage. From 1980 to 2020, the regulating services, cultural services, hydrological regulation, and waste disposal all showed good upward trends, while all other secondary services showed decreasing trends (Table 6). Among them, the most significant decrease of 6.2% was observed for the soil maintenance function, and the most extensive enhancement of 51.6% was observed for the regulation service function. These were closely related to the implementation of the policy of returning farmland to lakes and the encroachment of urban construction on farmland and lake surfaces.

ESV in the study area was divided into five classes: low value [0, 14,221.07], lower value (14,221.07, 19,839.32], medium value (19,839.32, 25,117.07], higher value (25,117.07, 31,568.58], and high value (31,568.58, ∞) (CNY/ha) (Table 7). Regarding spatial distribution, the three regions in the eastern, southern, and northern parts of WUA had a low ecosystem service value rating, and the western Jianghan Plain had a higher rating (Figure 5). Among them, the high-value area was mainly concentrated in the Liangzi Lake, Rangdu Lake, and Dongjia Lake areas in Wuhan, the Tiaocha Lake area in Xiaogan, the Paihu Lake area in Xiantao, the northern part of Xiaogan, the eastern part of Huanggang, Huangshi, and the southern part of Xianning, etc. The excellent forest and lake resources created a relatively high regional ESV. Higher-value areas surrounded high-value areas, mainly in the northeast and southeast of the urban area. The mid-value areas surrounded the higher-value areas mainly in the peripheral areas of Wuhan; the lower-value and low-value areas were mainly in the western Jianghan Plain region, which is known for its farmland resources, lack of forest and water resources, and relatively low ESV overall. Regarding ESV changes, the ecological advantages of the northern part of WUA gradually waned (Figure 6). The ESV of Tongshan County of Xianning City, Yangxin County of Huangshi City, Laotian County of Huanggang City, and Yingshan County, which were famous for their forest ecological resources, gradually generated losses. Along with the acceleration of the co-location process, the ESV in Wuhan City and other urban areas showed a rapid decline.

#### 3.1.3. Ecological Management Recommendations

The rapid development of WUA has led to the continuous compression and erosion of “ecological space”, and the ecosystem is under stress. In terms of LER, the LER levels of several urban areas show an increasing and then decreasing development, with the most apparent change in the central area of Wuhan. However, with the expansion of the same urbanization, the region shows multiple core patches of ecological risk reduction. The area of higher- and high-risk areas in the urban area still show an expansion trend, and WUA still shows a depressed state regarding ecological risk control of the urban area on a region-wide scale. In terms of ESV, the high-value and higher-value areas of ESV in the urban area have increased. The ESV and water quality in several lakes have improved, and the function of water bodies has been gradually restored. However, several high-value independent ecological patches are surrounded by low-value areas, such as the Liangzi Lake, Rangdu Lake, and Dabie Mountain system ecological patches. The necessary ecological corridors are missing between the patches to establish ecological links.

Strengthening land use control and enhancing ecological service functions are necessary to guide the function of ecological land in the core area of urban development in the high-value ecological risk area, and ecological space should be rigidly controlled to regulate ecological service functions. Combined with lakes, wetlands, forests, and other ecological zones and wedge-shaped green areas around the periphery of Wuhan City in distant suburban areas, a regionally integrated ecological security framework needs to be constructed to clarify the boundaries of urban expansion and ecological protection, forming a framework for constructing ecological security in urban areas.

For the relatively independent high-value areas within each city in the city circle, emphasizing spatial planning and strengthening regional linkages are essential to develop an ecological safety pattern system for the whole metropolitan area in an integrated manner and connect the two obvious high-value agglomerations in the Makufu Mountains in the southeast and the Dabie Mountains in the northeast through green corridor spreading and greenways. At the same time, the ecological patches in the hinterland of the Jianghan Plain in the west need to be spatially linked to other high-value areas, and wetlands and lakes need to be restored to build an ecological network system.

### 3.2. Critical Area Identification and ESPs Construction

#### 3.2.1. Ecological Source Extraction

The ecological sources of WUA are mainly forest-type and lake-type ecological sources, which are determined by the geographic location of the Dabie Mountain System, Mofu Mountain System, Jianghan Plain, Yangtze River, Liangzi Lake, and Fucai Lake. Combining with the actual situation of the region, in order to avoid the influence of good patches, we comprehensively considered the size of WUA habitat patches, spatial distribution characteristics, and biodiversity and identified forest land with an area larger than 160 km^2^ and core patches of water bodies with an area larger than 70 km^2^ as the ecological source. Finally, we identified and extracted a total of 18 mountain forest ecological sources with an area of about 11,248.6 km^2^, accounting for 19.46% of the total area of the study area, and 12 water source ecological sources with an area of about 3125.4 km^2^, accounting for 5.41% of the total area of the study area.

Figure 7 shows that the overall distribution of WUA ecological sources was less uniform, with more full and prominent sites in the northeast and northwest and small and scattered sites in the west. Specifically, the forest ecological source was mainly distributed in the Dabie Mountain area in Laotian County, Yingshan County, and Herba County in the eastern part of Huanggang City, and the Makufu Mountain area in Chongyang County and Tongshan County in the southern part of Xianning City, of which the most extensive mountain forest ecological source was 3765.7 km^2^ in area. Overall, there were 30 ecological sources in WUA, the ecological source with the largest span was the Yangtze River and its tributaries, and the ecological source with the most significant area was the Makufu Mountain area. The ecological source of WUA as a whole was more dispersed, with significant differences in scale. The ecological source was subject to a large degree of human interference, which was not conducive to the flow of materials and energy between the ecological sources.

#### 3.2.2. Resistance Surface Construction

The natural environment and human activities significantly influence the resistance value, which can reflect the difficulty of various ecological sources in crossing the landscape. We used ArcGIS to analyze and calculate the resistance values of nine factors in the “ecology–nature–society” system and generated a single-factor resistance surface. Figure 8 showed that the places with high resistance values were mostly urbanized areas and areas along roads with low LER, ESV, and vegetation cover and high population density and land use intensity. The frequent human activities and ecological damage in these areas were not conducive to ecological processes and urgently need ecological restoration and protection. Then we used the raster calculator to weigh the calculation to obtain the comprehensive resistance surface (Figure 9). We can see that the high-value areas were mainly distributed in Wuhan City and various other urban areas with high human activity interference; the medium-value area distribution was widely distributed throughout the whole area, showing a trend of radiating reduction from WUA to the surrounding areas; the low-value areas were distributed in various ecological source areas, mainly concentrated in the southeast and northeast, where there is little construction and development and natural resources are abundant in mountainous forest areas.

#### 3.2.3. Ecological Corridor and Node Extraction

Ecological corridors are the minor resistance channels connecting two adjacent ecological source sites. In contrast, ecological nodes are located in the ecological corridors where ecological functions are more vulnerable and play a vital role in the operation of ecological flows. Based on the analysis of ecological sources, minimum resistance surfaces, and spatial and temporal changes in LER and ESV, this study constructed a total of 24 ecological corridors in WUA, graded according to their length (Figure 10). Three ecological corridors, namely, mountain forest, river, and composite, dominated WUA. Among them, the mountain-forest ecological corridor connects the natural landscape of the Mofu Mountain System and the Dabie Mountain System, maintaining a stable plant community and high biodiversity. River ecological corridors take the Yangtze River system as a carrier, build a buffer zone between the Makufu Mountain system, the Yangtze River system, and the Dabie Mountain system, control the urban expansion boundary, strengthen the construction of water connotation capacity, and improve the function of ecological river corridors and the ecological protection and management capacity of rivers and lakes. The composite ecological corridor enhances landscape heterogeneity, reduces lake water pollution, and organically combines the functions of water connotation, biological habitat, ecological agriculture, and cultural tourism. It provides a significant linkage to promote the construction of an ecological corridor network and an essential linkage for ecological activities such as material flow, energy cycle, biological migration, and species dispersal of the ecosystem in WUA.

In this study, a total of 42 ecological nodes was identified and graded according to the level of the corridor they were located in. The ecological nodes are mainly located in Wuhan City and the northeastern and southern mountain systems, with a spatial distribution pattern of “more in the east and less in the west”.

The nodes in Lake Diaoyu, the Han River basin, and Paihu are vital in the ecological construction of Xiaogan City, Xiantao City, Qianjiang City, and Tianmen City. The Jianghan Plain is surrounded by few ecological sources of mountains, forests, and water and lacks meaningful connections between ecological sources in the urban area. Therefore, we should increase the efforts to return the watershed to the lake, reduce the pollution from the perimeter farming, restore and protect the original lake wetland ecosystem, and help form a composite ecological corridor of mountains, forests, and lakes to provide a complete channel and medium for ecological flow. The ecological core nodes located in the essential mountainous areas in the southeast and northeast and around Wuhan City should strengthen the protection and restoration of wetland, mountain, and forest ecological functions, strictly prohibit the expansion of urban construction land, actively transform traditional industries into ecological agriculture, comprehensively enhance the ecosystem service functions, promote the continuous improvement of the regional ecological environment, and strengthen the connectivity of the ecological network in WUA. Meanwhile, forest panels or native landscape patches are added in critical areas located in several transition zones between Wuhan City and neighboring cities, such as Shuangfeng Mountain National Forest Park, Lifting Water Watershed, Liangzi Lake Watershed, and Wild Boar Lake Watershed, to control the expansion rate of surrounding construction land and introduce plants with strong pollution resistance to prevent the spread of agricultural non-point-source pollution.

## 4. Discussion

### 4.1. Methodological Advantages

The construction of ESPs is one of the fundamental ways to effectively guarantee the full utilization of regional ecosystem functions and services. Most previous studies have focused on using spatial relationships of grid elements and networks, with insufficient emphasis on landscape ecological processes and unclear ecological significance [64,65,66]. This study combined the development characteristics of WUA, reflected the characteristics of ecological processes through the dynamic changes in ESV and LER in WUA in 40 years, and constructed a landscape pattern resistance model from three aspects: ecological elements (LER, ESV), natural elements, and human elements, which make the construction of ecological corridors more targeted and thus improved the construction method of ESPs. Therefore, the research method here can be applied to future ecological governance, allowing for a more reasonable construction of ESPs.

### 4.2. Optimization Suggestions

Through research and analysis, we found that the ESV of WUA has shown a decreasing trend over the years, and urbanized areas and Jianghan Plain areas were to be significantly lower; the expansion of LER grade and area also accompanied the expansion of urban construction land. As the core area of the riverine urban zone in the middle and lower reaches of the Yangtze River, the continuous accelerated urbanization expansion of WUA has caused severe negative impacts on it and its surrounding areas in terms of declining ecosystem service functions and increasing ecological risks. We found that the low-resistance areas in the Makufu and Dabie Mountain systems have good ecological resources and high forest areas. In contrast, the high-resistance areas were mainly located in the Jianghan Plain area and the construction land catchment area. This is because the weak natural environment and human activities have significantly changed the landscape pattern of the land surface, and urban expansion and industrial activities have caused significant disturbance to the surrounding ecological environment. As for the Jianghan Plain region, an essential base for agricultural production in China, its rough land use pattern has led to severe pollution of lake wetlands, shrinkage of wetland areas, and degradation of ecosystem functions.

Based on the above study, considering the spatial and temporal changes of LER and ESV and the resistance surface model, we have identified 30 ecological sources, constructed 24 ecological corridors and 42 ecological nodes in WUA, and built a multi-level ecological network of “point–line–surface” to realize the effective linkage of rivers, lakes, plains, and mountains, and to promote the circulation of the ecosystem structure network, to provide a reference for the establishment of an ecological and environmental cooperative governance mechanism and ecosystem planning in WUA. First of all, we can increase the protection of ecological source sites and various parks with specific vegetation coverage, continue to implement the ecological restoration and protection of lakes and comprehensive upgrading projects, delineate ecological red lines, and strictly implement the successful completion of the return of lakes and ecological restoration work in WUA [67]. At the same time, radiating ecological safety buffer zones on the periphery of ecological source sites serve as a transition zone between ecological source sites and urban construction development zones. For example, buffer zones are established in ecological source areas around urban areas in Wuhan and other cities to implement ecological hierarchical management and differential control actively and assume a barrier role. Furthermore, the Jianghan Plain area should carry out wetland protection and restoration, agricultural ecological protection, actively coordination of the relationship between flood control and storage and development of agriculture, removal of obstacles to river flooding, promotion of the transformation of conservation and intensive agriculture, and exploration of regional agricultural sustainable development strategies [68]. Moreover, the cultivation of ecological nodes should be increased, and the protection and restoration of important corridors should be strengthened. The study area should focus on cultivating ecological nodes such as mountain and forest land, ponds and reservoirs, and water source protection areas for exemplary management. Gradually, some of the strip and belt ecological corridors will be widened, and ecological corridor buffers will be set to enhance ecological effects. For example, the construction of the Yangtze River ecological corridor should integrate its flood control capacity, improve ecological restoration along the river, build a crucial ecological barrier along the Yangtze River, and reduce ecological risks. Several ecological corridors in Wuhan closely link various lakes and wetlands to form an urban ecological belt, improve the quality and stability of ecosystem services, give play to its ecological effectiveness in taking over neighboring areas, and guarantee the efficient operation of the ecological network [69].

### 4.3. Limitations of the Study

Although the inclusion of ESVs and LERs into the ecological safety evaluation system in this study helps to optimize the construction model of ESPs, there is still space for further improvement and expansion of this study. For example, based on previous studies, this study only provides judgment for ecological source identification. In addition, the landscape resistance model identified ecological corridors and did not set the width of the corridors, which affected the ecological functions and the types of migrating species. Therefore, in future studies, detailed investigation of ecological source sites and research methods to explore corridor width identification should be increased.

## 5. Conclusions

WUA assumes an essential ecological function for improving the ecological spatial system along the river basin. At the same time, the subtle changes in its ecosystem have a significant impact on the ecological security of the Yangtze River basin. For a long time, the rapid urbanization and development of WUA has brought about the deterioration of environmental quality, urban haze pollution, lake pollution, and other problems, while the LER and ESV have also undergone drastic changes, and the ecological quality and safety of the urban agglomerations urgently needs to be improved. The analysis from the perspective of WUA can maintain the integrity and coordination of the ecosystem on a large scale and better explore the ecological security pattern of synergistic protection of the ecological environment between WUA and the middle and lower reaches of the Yangtze River basin. This paper analyzes the dynamic changes in ESV and LER of WUA from 1980 to 2020 based on land use data. It constructs an “ecology–nature–human” evaluation system by incorporating ESV and LER to derive the ecological security pattern of WUA. As a new research perspective, method, and practical path to building ESPs, this paper emphasizes the spatial characteristics of and spatial and temporal changes in ecological processes, further enhancing the relevance of LER and ESV. The research results provide references for the construction of ESPs and optimization of urban development layout in WUA.

## Figures and Tables

**Figure 1 ijerph-20-02792-f001:**
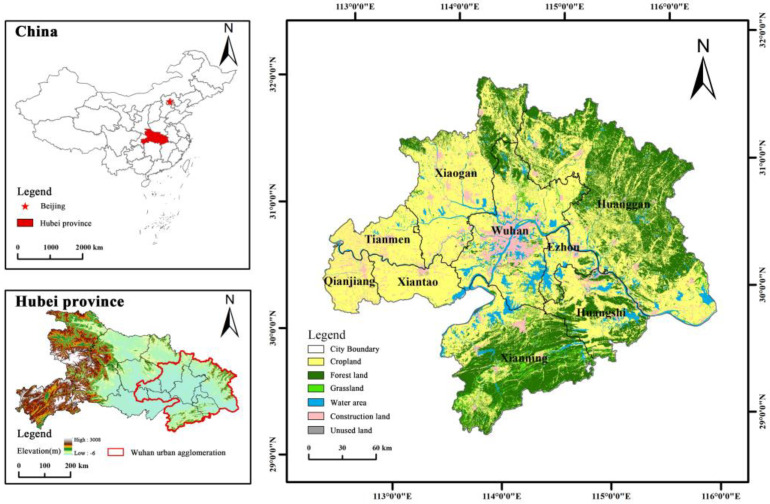
Location of the study area.

**Figure 2 ijerph-20-02792-f002:**
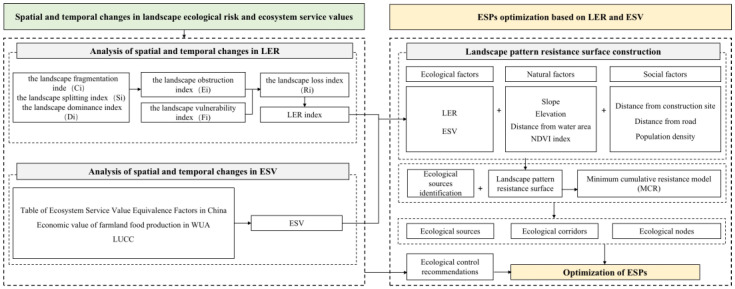
Research ideas and framework.

**Figure 3 ijerph-20-02792-f003:**
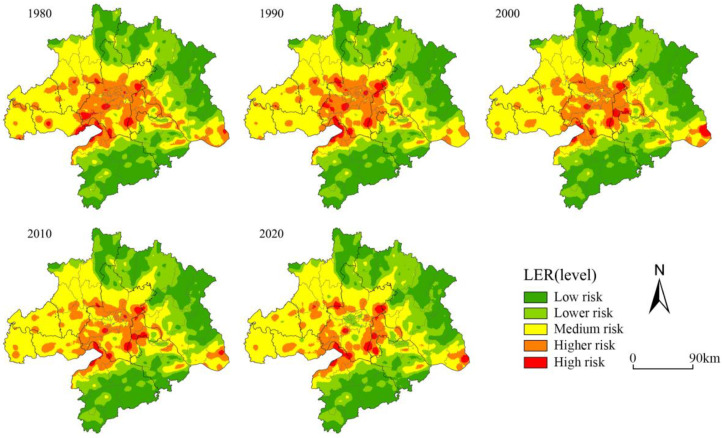
Spatial distribution pattern of landscape ecological risk from 1980 to 2020.

**Figure 4 ijerph-20-02792-f004:**
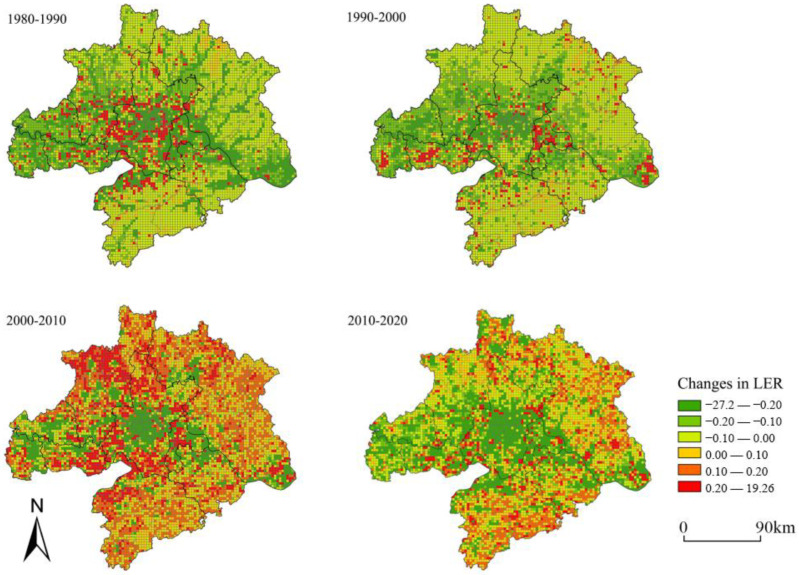
Change in landscape ecological risk from 1980 to 2020.

**Figure 5 ijerph-20-02792-f005:**
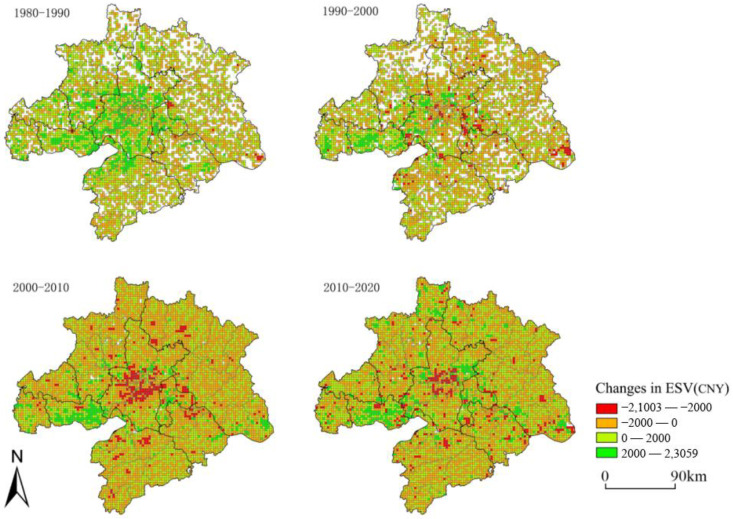
Spatial distribution pattern of ecosystem service value from 1980 to 2020.

**Figure 6 ijerph-20-02792-f006:**
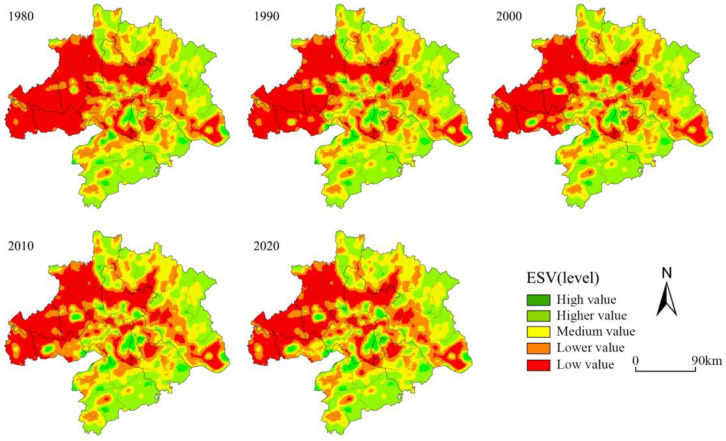
Change in ecosystem service value from 1980 to 2020.

**Figure 7 ijerph-20-02792-f007:**
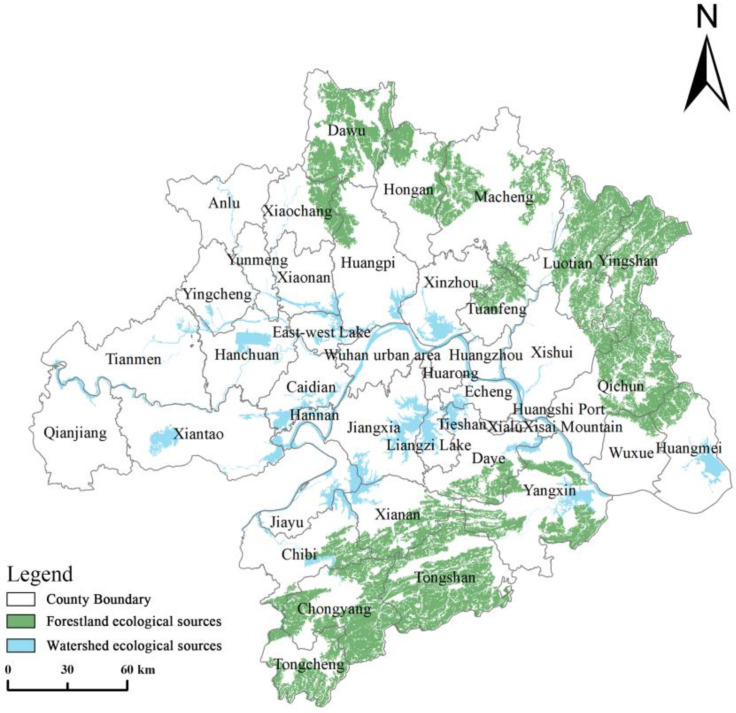
Ecological sources of WUA.

**Figure 8 ijerph-20-02792-f008:**
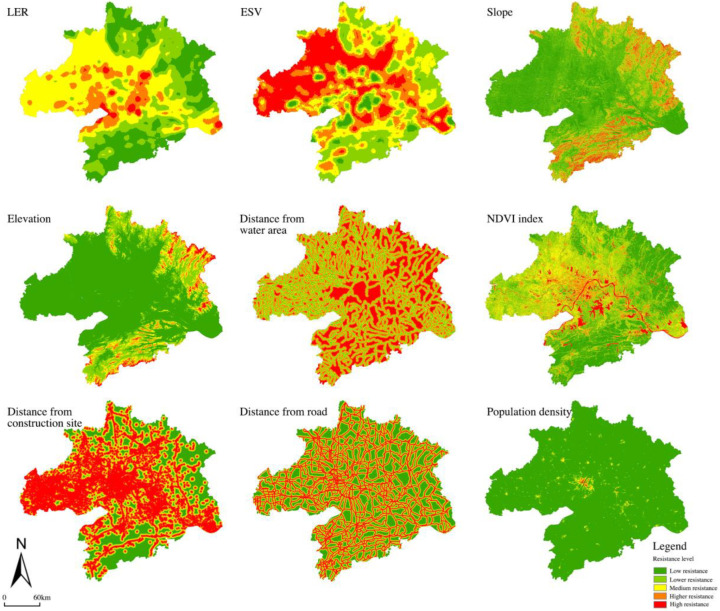
Spatial distribution of ecological resistance factors.

**Figure 9 ijerph-20-02792-f009:**
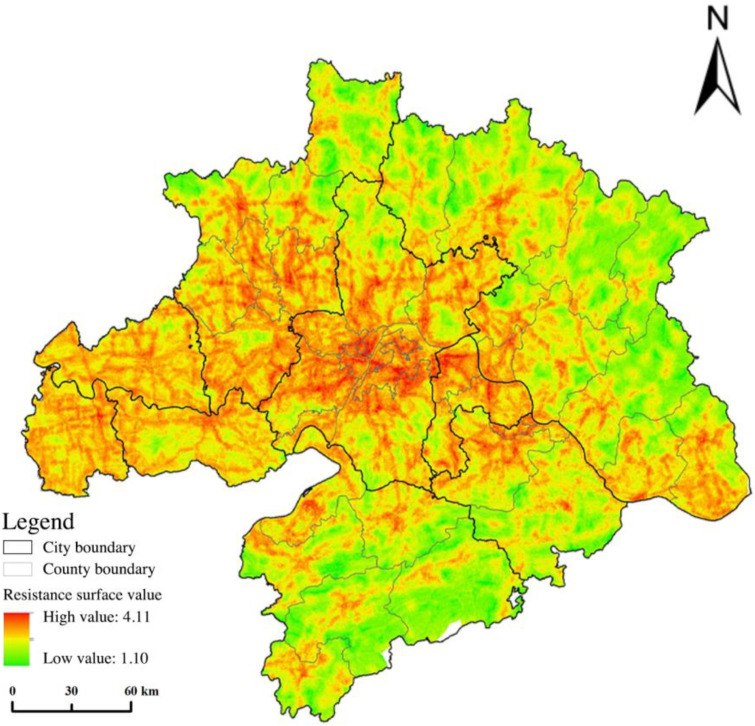
The WUA ecological resistance surface.

**Figure 10 ijerph-20-02792-f010:**
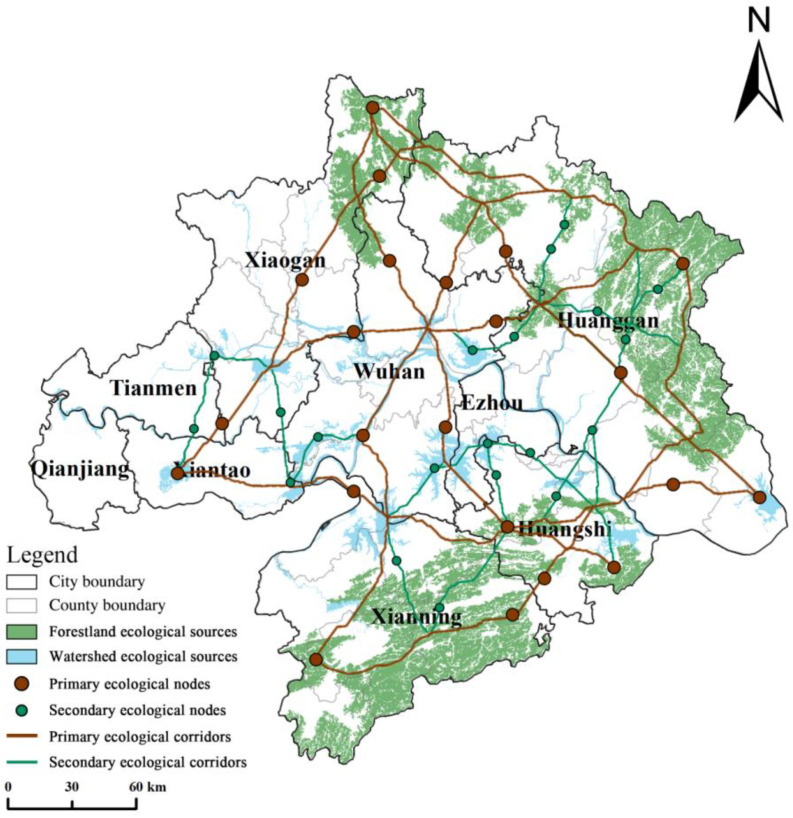
The WUA landscape pattern optimization network.

**Table 1 ijerph-20-02792-t001:** Ecosystem service value per unit area of different terrestrial ecosystem types (CNY·hm^−2^).

Primary Type	Secondary Type	Farmland	Forest	Grassland	Water	Unused Land
Provisioning services	Food production	1146.22	378.25	492.87	607.50	22.92
	Raw material production	447.03	3415.74	412.64	401.18	45.85
Regulating services	Gas regulation	825.28	4951.67	1719.33	584.57	68.77
	Climate regulation	1111.83	4665.12	1788.10	2361.21	149.01
	Hydrological adjustment	882.59	4688.04	1742.25	21,514.55	80.24
	Waste treatment	1593.25	1971.50	1513.01	17,021.37	298.02
Supporting services	Soil conservation	1684.94	4607.80	2567.53	469.95	194.86
	Biodiversity maintenance	1169.14	5169.45	2143.43	3931.53	458.49
Cultural services	Aesthetic landscape	194.86	2384.14	997.21	5089.22	275.09

**Table 2 ijerph-20-02792-t002:** Ecological resistance index and grading evaluation criteria.

Evaluation Dimension	Evaluation Indicator	Resistance Classification					Weights
		1	2	3	4	5	
Ecological factors	LER	0.0067~0.0010	0.0010~0.0126	0.0126~0.0153	0.0153~0.0196	0.0196~0.0329	0.1119
	ESV	31,585.0~47,490.2	25,086.1~31,585.0	19,784.4~25,086.1	14,140.6~19,784.4	3879.2~14,140.6	0.1248
Natural Factors	Slope/(°)	0~4.4	4.4~10.2	10.2~17.8	17.8~27.2	>27.2	0.0771
	Elevation/m	<89	89~227	227~419	419~695	695~1727	0.0751
	Distance from water bodies/m	<100	100~500	500~1000	1000~2500	>2500	0.1133
	NDVI	0.601~0.892	0.459~0.601	0.301~0.459	0.031~0.301	−0.200~0.031	0.1387
Social factors	Distance from construction land/m	>4000	3000~4000	2000~3000	1000~2000	<1000	0.1594
	Distance from roads/m	>2500	1500~2500	1000–1500	500~1000	<500	0.1009
	Population density/(person/km^2^)	0~24.9	24.9~116.1	116.1~273.8	273.8~597.3	597.3~2115.4	0.0988

**Table 3 ijerph-20-02792-t003:** Ecological risk index of WUA, 1980–2020.

Landscape Type	Time	Area (ha)	Number of Patches (pcs)	Fragmentation Index (*Ci*)	Splitting Index (*Si*)	Dominance Index (*Di*)	Obstruction Index (*Ei*)	Loss Index (*Ri*)
Farmland	1980	3,125,775.1	7326	0.00234	0.03295	0.31816	0.07469	0.01419
	1990	3,031,543.4	7535	0.00249	0.03446	0.31386	0.07435	0.01413
	2000	2,990,827.6	7560	0.00253	0.03499	0.31066	0.07389	0.01404
	2010	2,844,254.3	9317	0.00328	0.04084	0.31457	0.07680	0.01459
	2020	2,765,090.6	9600	0.00347	0.04264	0.31286	0.07710	0.01465
Forest	1980	1,760,203.5	6596	0.00375	0.05553	0.21367	0.06127	0.00613
	1990	1,760,254.8	6598	0.00375	0.05553	0.21307	0.06115	0.00611
	2000	1,754,703.3	6630	0.00378	0.05584	0.21249	0.06114	0.00611
	2010	1,740,329.2	7403	0.00425	0.05950	0.21407	0.06279	0.00628
	2020	1,744,336.6	7720	0.00443	0.06061	0.21849	0.06409	0.00641
Grassland	1980	145,210.2	1593	0.01097	0.33077	0.03228	0.11117	0.01556
	1990	144,709.5	1600	0.01106	0.33264	0.03219	0.11176	0.01565
	2000	144,458.3	1610	0.01115	0.33426	0.03215	0.11228	0.01572
	2010	141,131.3	1676	0.01188	0.34908	0.03100	0.11686	0.01636
	2020	140,315.4	1647	0.01174	0.34801	0.03060	0.11639	0.01630
Water	1980	463,029.8	5859	0.01265	0.19894	0.11381	0.08877	0.02130
	1990	555,504.1	6042	0.01088	0.16839	0.12217	0.08039	0.01929
	2000	571,683.1	6185	0.01082	0.16555	0.12467	0.08001	0.01920
	2010	606,615.5	6730	0.01109	0.16275	0.12725	0.07982	0.01916
	2020	63,3003.6	6512	0.01029	0.15339	0.12639	0.07644	0.01835
Construction Land	1980	267,245.1	20,964	0.07844	0.65199	0.31126	0.29707	0.01485
	1990	283,638.0	20,949	0.07386	0.61409	0.31019	0.28319	0.01416
	2000	306,838.4	21,000	0.06844	0.56835	0.31045	0.26681	0.01334
	2010	440,785.4	21,556	0.04890	0.40084	0.30383	0.20547	0.01027
	2020	491,254.6	21,277	0.04331	0.35728	0.30405	0.18965	0.00948
Unused land	1980	31,343.8	620	0.01978	0.95600	0.01082	0.29886	0.08368
	1990	17,122.5	529	0.03090	1.61649	0.00852	0.50210	0.14059
	2000	24,277.2	574	0.02364	1.18760	0.00958	0.37002	0.10361
	2010	19,667.3	625	0.03178	1.52971	0.00929	0.47666	0.13346
	2020	17,252.8	505	0.02927	1.56726	0.00760	0.48633	0.13617

**Table 4 ijerph-20-02792-t004:** Distribution of landscape ecological risk areas from 1980 to 2020.

Landscape Ecological Risk Level	1980		1990		2000		2010		2020	
	Area/km^2^	Percentage/%	Area/km^2^	Percentage/%	Area/km^2^	Percentage/%	Area/km^2^	Percentage/%	Area/km^2^	Percentage/%
Low	14,919.3	25.74	15,431.94	26.62	15,596.28	26.90	15,519.78	26.77	14,435.1	24.90
Lower	12,359.61	21.32	12,292.11	21.20	12,555.45	21.66	12,432.6	21.45	13,804.47	23.81
Medium	19,502.55	33.64	20,722.23	35.75	20,737.71	35.77	20,833.92	35.94	21,881.61	37.75
Higher	10,031.13	17.30	8225.19	14.19	8102.88	13.98	8251.11	14.23	6671.07	11.51
High	1159.2	2.00	1299.51	2.24	1011.15	1.74	932.85	1.61	1161.54	2.00

**Table 5 ijerph-20-02792-t005:** Changes in ecosystem service value of the six land cover types during 1980–2020.

Landscape Type	ESV (×10^8^ CNY)					ESV Change Rate (%)			
	1980	1990	2000	2010	2020	1980–1990	1990–2000	2000–2010	2010–2020
Farmland	283.043	274.51	270.824	257.551	250.383	−3.015	−1.343	−4.901	−2.783
Forest	567.344	567.36	565.571	560.938	562.23	0.003	−0.315	−0.819	0.23
Grassland	19.424	19.357	19.323	18.878	18.769	−0.345	−0.174	−2.303	−0.578
Water	240.688	288.757	297.167	315.325	329.042	19.972	2.912	6.11	4.35
Unused land	0.499	0.273	0.387	0.313	0.275	−45.372	41.785	−18.989	−12.277
Total	1110.998	1150.257	1153.272	1153.006	1160.698	3.534	0.262	−0.023	0.667

**Table 6 ijerph-20-02792-t006:** Change in each ecosystem service value during 1980–2020.

Service Type	ESV (×10^8^ CNY)					ESV Change Rate (%)			
	1980	1990	2000	2010	2020	1980–1990	1990–2000	2000–2010	2010–2020
Provisioning services	122.590	122.009	121.316	118.754	117.902	−0.474	−0.568	−2.111	−0.718
Regulating services	626.515	660.606	664.643	670.073	678.109	5.441	0.611	0.817	1.199
Supporting services	288.737	290.007	289.049	284.811	284.050	0.440	−0.330	−1.466	−0.267
Cultural services	73.155	77.635	78.264	79.368	80.637	6.124	0.810	1.410	1.599
Food production	46.022	45.498	45.109	43.569	42.833	−1.138	−0.855	−3.413	−1.690
Raw material production	76.568	76.511	76.207	75.185	75.069	−0.075	−0.398	−1.341	−0.154
Gas regulation	118.181	117.928	117.412	115.635	115.318	−0.214	−0.437	−1.514	−0.274
Climate regulation	130.445	131.553	131.230	129.688	129.600	0.849	−0.246	−1.175	−0.068
Hydrological regulation	212.281	231.327	234.189	239.676	244.826	8.972	1.237	2.343	2.149
Waste treatment	165.608	179.798	181.812	185.075	188.365	8.568	1.120	1.795	1.778
Soil conservation	139.739	138.548	137.690	134.628	133.577	−0.853	−0.620	−2.224	−0.781
Biodiversity maintenance	148.998	151.459	151.359	150.183	150.474	1.651	−0.066	−0.777	0.193
Aesthetic landscape	73.155	77.635	78.264	79.368	80.637	6.124	0.810	1.410	1.599

**Table 7 ijerph-20-02792-t007:** Distribution of the ecosystem service value areas from 1980 to 2020.

ESV	1980		1990		2000		2010		2020	
	Area/km^2^	Percentage/%	Area/km^2^	Percentage/%	Area/km^2^	Percentage/%	Area/km^2^	Percentage/%	Area/km^2^	Percentage/%
Low	17,522.82	30.24	15,784.38	27.24	15,334.29	26.47	14,855.04	25.64	14,401.17	24.86
Lower	12,393.18	21.39	11,803.68	20.37	12,211.02	21.08	13,035.33	22.50	13,047.84	22.52
Medium	14,086.62	24.31	15,062.04	26.00	15,184.89	26.21	15,251.94	26.33	15,412.23	26.60
Higher	13,048.47	22.52	13,889.16	23.97	13,873.14	23.95	13,266.72	22.90	13,371.93	23.08
High	885.96	1.53	1397.79	2.41	1333.71	2.30	1528.02	2.64	1703.88	2.94

## Data Availability

Not applicable.

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
