# Peer review of "Integrating Landscape Ecological Risks and Ecosystem Service Values into the Ecological Security Pattern Identification of Wuhan Urban Agglomeration"

_ijerph, 2023, doi:10.3390/ijerph20042792_

Round 1
Reviewer 1 Report
The analysis examines the problem based on a thorough, well-planned research plan. Although the methodology and results are clearly formulated, the research question should be formulated more plastically. The research question is not presented clearly enough either in the abstract or in the introduction. In accordance with this, the conclusion should be explained better, it currently gives a too concise summary.
From a formal point of view, the graphic quality of the map representations should be highlighted, diagrams help the reader to understand. At the same time, the text still needs to be reviewed linguistically. There are typos in several places in the text, mainly spaces are missing.
Author Response
Response to Reviewer 1 Comments
Dear reviewer:
Thank you for your decision and constructive comments on my manuscript. We have carefully considered the suggestion of Reviewer and make some changes. We have tried our best to improve and made some changes in the manuscript.
Revision notes, point-to-point, are given as follows:
Point 1: The analysis examines the problem based on a thorough, well-planned research plan. Although the methodology and results are clearly formulated, the research question should be formulated more plastically. The research question is not presented clearly enough either in the abstract or in the introduction. In accordance with this, the conclusion should be explained better, it currently gives a too concise summary.
Response 1: Thank you for your suggestions regarding the lack of description of the research questions. In the abstract and introduction of the paper, we have re-emphasized the research problem with the regional ecological safety of the urban area as the core. The regional ecological issues of the current problems of the WUA are supplemented by emphasizing the impact of rapid urbanization on the regional ecology and the urgent need to address it (Line 16 and 100). In the conclusion, we also highlight the research questions more prominently, giving a more detailed explanation(Line 634).
Point 2: From a formal point of view, the graphic quality of the map representations should be highlighted, diagrams help the reader to understand. At the same time, the text still needs to be reviewed linguistically. There are typos in several places in the text, mainly spaces are missing.
Response 2: We apologize for the poor language of our manuscript and quality of the graphics. We worked on the manuscript for a long time, and repeatedly adding and removing sentences and chapters apparently led to poor readability. We have now worked on both language and readability, and have also involved native English speakers to correct the language. We really hope to make substantial improvements in both fluency and language level. At the same time, we have improved the quality of several graphics and improved the graphical presentation to help the reader understand the paper better.
Reviewer 2 Report
Thank you for giving me this opportunity to read the manuscript entitled "Integrating landscape ecological risks and ecosystem service values into the ecological security pattern identification of Wuhan Urban Agglomeration". The topic of this manuscript is interesting and would be a good contribution to this field. I think it could be considered for publication in IJERPH once the following issues are addressed.
1. Please replace the keywords that already appear in the manuscript's title with close synonyms or other keywords, which will also facilitate your paper being searched by potential readers.
2. Please enlarge the text in the Figures to make them read clearly.
3. Line 58: The full name of “UN” should be given when it first appear in the manuscript.
4. Line 135: “accuracy rate of each land type is over 90%...” a reference should be provided to support the statement.
5. Line 140: More detailed information regarding the data collection of DEM should be provided.
6. Line 41: “… degradation problems[1-3].”: a paper titled “How does urban expansion impact people’s exposure to green environments? A comparative study of 290 Chinese cities” is suggested to be added as a reference to support the statement.
7. Line 57-58: “in human well-being, health, and the interdependence of humans and all life on earth[13-16].”: a paper titled “Observed inequality in urban greenspace exposure in China” could also be used as a reference.
8. Limitation section should be added as a sub-section to the Discussion.
9. Some grammatical errors exist in the manuscript. Therefore, a critical review of the manuscript's language will improve its readability.
Author Response
Response to Reviewer 2 Comments
Dear reviewer:
Thank you for your praise and constructive comments on my manuscript. We have carefully considered the suggestions of the reviewer and made some changes. We have tried our best to improve and make some changes to the manuscript.
Revision notes, point-to-point, are given as follows:
Point 1: Please replace the keywords that already appear in the manuscript's title with close synonyms or other keywords, which will also facilitate your paper being searched by potential readers.
Response 1: We have replaced keywords with near-synonyms or other keywords in a timely manner, which provides assistance to potential readers in their search.
Point 2: Please enlarge the text in the Figures to make them read clearly.
Response 2: We have enlarged the text in the study frame as well as the text in the other figures, which will be better readable.
Point 3: Line 58: The full name of “UN” should be given when it first appear in the manuscript.
Response 3: We have revised the first occurrence of "UN" to "United Nations".
Point 4: Line 135: “accuracy rate of each land type is over 90%...” a reference should be provided to support the statement.
Response 4: We have cited a paper titled "Spatial-temporal evolution of land use and ecological risk in Dongting Lake basin during 1980-2018" as proof that "The accuracy of each land type is more than 90% ....." .
Point 5: Line 140: More detailed information regarding the data collection of DEM should be provided.
Response 5: In line 150, we have highlighted that the DEM data come from the Resource and Environment Data Cloud platform of the Institute of Geographical Sciences and Resources, Chinese Academy of Sciences.
Point 6: Line 41: “… degradation problems[1-3].”: a paper titled “How does urban expansion impact people’s exposure to green environments? A comparative study of 290 Chinese cities” is suggested to be added as a reference to support the statement.
Response 6: Thank you for your suggestions on the references. We have introduced a paper entitled "How does urban expansion impact people's exposure to green environments? A comparative study of 290 Chinese cities" in "... upgrading problems [1-3].".
Point 7: Line 57-58: “in human well-being, health, and the interdependence of humans and all life on earth[13-16].”: a paper titled “Observed inequality in urban greenspace exposure in China” could also be used as a reference.
Response 7: Thank you for your suggestions on the references. We have introduced a paper entitled "Observed inequality in urban green space exposure in China" in "human well-being, health, and the interdependence of humans and all life on earth [13-16].".
Point 8: Limitation section should be added as a sub-section to the Discussion.
Response 8: Thank you for your suggestions on Discussion. In conjunction with the structure of the lines in Discussion, we have included the limitations of the paper as a separate subsection.
Point 9: Some grammatical errors exist in the manuscript. Therefore, a critical review of the manuscript's language will improve its readability.
Response 9: We apologize for the poor language of our manuscript. We worked on the manuscript for a long time, and repeatedly adding and removing sentences and chapters apparently led to poor readability. We have now worked on both language and readability, and have also involved native English speakers to correct the language. We really hope to make substantial improvements in both fluency and language level.